# Comparative Proteomic Analysis of Secretory Proteins of *Mycoplasma bovis* and *Mycoplasma mycoides* subsp. *mycoides* Investigates Virulence and Discovers Important Diagnostic Biomarkers

**DOI:** 10.3390/vetsci10120685

**Published:** 2023-12-01

**Authors:** Ali Sobhy Dawood, Gang Zhao, Yujia He, Doukun Lu, Shujuan Wang, Hui Zhang, Yingyu Chen, Changmin Hu, Huanchun Chen, Elise Schieck, Aizhen Guo

**Affiliations:** 1National Key Laboratory of Agricultural Microbiology, Huazhong Agricultural University, Wuhan 430070, China; ali.dawood@vet.usc.edu.eg (A.S.D.); hyj980929@sina.com (Y.H.); doukunlu@webmail.hzau.edu.cn (D.L.); shujuanwwang@sina.com (S.W.); 80300228@swun.edu.cn (H.Z.); chenyingyu@mail.hzau.edu.cn (Y.C.); hcm@mail.hzau.edu.cn (C.H.); chenhch@mail.hzau.edu.cn (H.C.); 2Hubei Hongshan Laboratory, Huazhong Agricultural University, Wuhan 430070, China; 3College of Veterinary Medicine, Huazhong Agricultural University, Wuhan 430070, China; 4Hubei International Scientific and Technological Cooperation Base of Veterinary Epidemiology, Huazhong Agricultural University, Wuhan 430070, China; 5International Research Center for Animal Disease, Ministry of Science and Technology, Huazhong Agricultural University, Wuhan 430070, China; 6Infectious Diseases, Faculty of Veterinary Medicine, University of Sadat City, Sadat City 32897, Egypt; 7Key Laboratory of Ministry of Education for Conservation and Utilization of Special Biological Resources in the Western China, School of Life Sciences, Ningxia University, Yinchuan 750021, China; zhaogang@nxu.edu.cn; 8International Livestock Research Institute, Nairobi 00100, Kenya

**Keywords:** *Mycoplasma bovis*, *Mycoplasma mycoides* subsp. *mycoides*, secreted proteins, proteomic analysis, diagnosis, virulence

## Abstract

**Simple Summary:**

*Mycoplasma bovis* (*M. bovis*) and *Mycoplasma mycoides* subsp. *mycoides* (*Mmm*) are the most important pathogenic mycoplasma species. But, the limited knowledge of the secreted proteins of *Mmm* and *M. bovis* has obstructed our understanding of mycoplasmal pathogenicity. In order to remove the interference of serum proteins, we used a serum-free medium to culture mycoplasma. The secreted proteins were extracted and identified by liquid chromatography-tandem mass spectrometry (LC-MS/MS). Further comparative proteomic analysis was performed to reveal homologous and unique secreted proteins and discover differential diagnostic biomarkers between *M. bovis* and *Mmm*. The findings are significant to further investigate the virulence or immunological properties of related secreted proteins for elucidating the pathogenesis and immune response of both mycoplasmas.

**Abstract:**

The most important pathogenic *Mycoplasma* species in bovines are *Mycoplasma bovis* (*M. bovis*) and *Mycoplasma mycoides* subsp. *mycoides* (*Mmm*). *Mmm* causes contagious bovine pleuropneumonia (CBPP), which is a severe respiratory disease widespread in sub-Saharan Africa but eradicated in several countries, including China. *M. bovis* is an important cause of the bovine respiratory disease complex (BRD), characterized worldwide by pneumonia, arthritis, and mastitis. Secreted proteins of bacteria are generally considered virulence factors because they can act as toxins, adhesins, and virulent enzymes in infection. Therefore, this study performed a comparative proteomic analysis of the secreted proteins of *M. bovis* and *Mmm* in order to find some virulence-related factors as well as discover differential diagnostic biomarkers for these bovine mycoplasmas. The secretome was extracted from both species, and liquid chromatography-tandem mass spectrometry was used, which revealed 55 unique secreted proteins of *M. bovis*, 44 unique secreted proteins of *Mmm*, and 4 homologous proteins. In the *M. bovis* secretome, 19 proteins were predicted to be virulence factors, while 4 putative virulence factors were identified in the *Mmm* secretome. In addition, five unique secreted proteins of *Mmm* were expressed and purified, and their antigenicity was confirmed by Western blotting assay and indirect ELISA. Among them, Ts1133 and Ts0085 were verified as potential candidates for distinguishing *Mmm* infection from *M. bovis* infection.

## 1. Introduction

Mycoplasmas are minimal and wall-less bacteria that include many pathogenic species that cause respiratory diseases, arthritis, and urogenital tract diseases in humans and animals [1]. *Mycoplasma bovis* (*M. bovis*) and *Mycoplasma mycoides* subsp. *mycoides* (*Mmm*) are two pathogenic species in bovines. *Mmm* is a well-known pathogen causing contagious bovine pleuropneumonia (CBPP), a notifiable disease listed in the World Organization for Animal Health (WOAH) characterized by pneumonia, pleurisy, and a high mortality rate of up to 50% without treatment [2]. *Mmm* has been eradicated in many parts of the world, including Europe (Portugal, Spain, France, and Italy) [3], Australia [4], and China [5], however, it is still the most serious animal disease endemic in vast areas of sub-Saharan Africa [6]. On the other hand, *M. bovis* is a common cause of subacute and chronic pneumonia in feedlot cattle and dairy calves throughout the world [7]. Due to the poor outcome of antibiotic treatment and the unsatisfactory efficacy of vaccines against CBPP and the lack of commercial vaccines available for *M. bovis*, both pathogens have gained significant economic importance for the worldwide cattle industry. The main reason for this status is attributed to a poor understanding of the pathogenesis and immune responses of both mycoplasma species [8].

Secreted proteins of bacteria are generally considered virulence factors because they usually function as toxins, adhesins, and virulence-determining enzymes, thereby mediating bacterial adhesion, invasion, propagation, and suppressing host defense [9]. Accordingly, several secreted proteins of mycoplasmas have been identified as virulence factors, including P102 of *M. hyopneumoniae* [10], Mpn491 and the community-acquired respiratory distress syndrome (CARDS) toxin of *M. pneumoniae* [11,12], and MBOV_RS02825 and MbovP280 of *M. bovis* [13,14]. Also, identifying novel immunogenic proteins for improved diagnostics and vaccines for *Mmm* and *M. bovis* has been confirmed [15,16]. Further, the secretome and extracellular vesicles of several mycoplasma species have been identified by liquid chromatography-tandem mass spectrometry (LC-MS/MS), isobaric tags for relative and absolute quantitation (iTRACK), and label-free proteomic analyses [17,18,19,20]. In addition, comparative proteomics has been used to analyze the whole-cell proteomes of two *Mycoplasma hyopneumoniae* strains [21], the whole-cell proteomes of the virulent *M. hyopneumoniae* strain and its attenuated strain [22], and the secretomes of *Mycoplasma hyopneumoniae* and *Mycoplasma flocculare* [19]. However, since mycoplasmas are fastidious species that require their media to contain sera, which could change the mycoplasmas protein pattern [23] and contaminate the extracted secreted proteins, the above studies could not exclude the interference of the foreign proteins from the sera of the medium. Therefore, serum-free media would be an ideal solution to avoid contamination of secreted proteomics of mycoplasmas by sera.

In the present study, a semi-defined medium without serum was used to culture *M. bovis* strain HB0801 and *Mmm* strain Afadé for the extraction of the soluble secreted proteins in the culture supernatant, and LC-MS/MS was used to identify secreted proteomics. The further comparative analysis was performed to first reveal homologous and unique secreted proteins between *M. bovis* and *Mmm*. The findings are significant to further investigate the virulence or immunological properties of related secreted proteins for elucidating the pathogenesis and immune response of both mycoplasmas.

## 2. Materials & Methods

### 2.1. Bacterial Strains and Culture

*Mycoplasma mycoides* subsp. *mycoides* strain Afadé (GenBank accession no. GCA_000959065.1) originates from Northern Cameroon and was isolated at the Farcha laboratories in Tchad in 1965 [2]. *M. bovis* strain HB0801 (GenBank accession no. NC_018077.1) was isolated and characterized by the National Key Laboratory of Agricultural Microbiology, HZAU, China [7]. *Mmm* was grown in the semi-defined medium without serum [24], while *M. bovis* was grown in the same semi-defined medium without serum but supplemented with sodium pyruvate (0.01 mol/L). The semi-defined medium was produced without animal serum or bovine serum albumin, as previously described [24]. In order to observe the growth of mycoplasmas, their growth curves were determined with the conventional colony counting method.

### 2.2. Enrichment of Proteins from Culture Supernatant of Mycoplasma

For enrichment of the soluble secreted proteins of *M. bovis* and *Mmm* from their culture supernatants, both were cultured in 500 mL semi-defined medium for 48 h at 37 °C. The bacteria were pelleted by centrifugation at 3500 g for 30 min, and culture supernatants were filtered through a 0.22 μm filter membrane. Then, the filtrates were centrifuged at 12,000 g for 30 min, filtered (0.22 μm), and dialyzed against sodium acetate at 5 mM and pH 5.0 using the SnakeSkin^®^ Dialysis Tubing (Thermo Scientific, IL, USA), with a molecular weight cut-off of 3.5 kDa. The final filtrates were freeze-dried and resolved in 10 mL PBS (pH 7.4).

### 2.3. Identification of Soluble Secreted Proteins by LC-MS/MS

A total of 100 μg of the secretome of *M. bovis* and *Mmm* was resolved with 12% SDS-PAGE. The gel samples for the secretomes of *M. bovis* and *Mmm* were prepared in parallel. Each sample was digested with 5 μL of 2.5–10 ng/μL trypsin solution (Promega, USA) at 37 °C for 20 h. Then, the solution was transferred into a new tube, and 100 μL of supersaturated alpha-cyano-4-hydroxycinnamic acid matrix solution (the solvent is 50% CAN and 0.1% TFA) were added, and the solvent was freeze-dried. In total, 60 μL of ddH_2_O (containing 0.1% formaldehyde) was used to resuspend the freeze-dried sample, and LC-MS/MS was performed by a previously described method [20]. Mascot 2.2 software was used for LC-MS/MS queries from NCBI (NCBI RefSeq assembly: *M. bovis* HB0801 (GCF_000270525.1), *Mmm* Afade (GCF_000959065.1)).

### 2.4. In Silico Comparative and Functional Analysis

In silico functional analysis of the *M. bovis* and *Mmm* secretomes was performed according to the previous description with minor modifications. Subcellular protein localization was predicted using PSORTb version 3.0.2 (https://www.psort.org/psortb/ accessed on 1 September 2022). Classically secreted proteins that carry signal sequences and are, therefore, secreted by classical pathways were predicted by the SignalP 5.0 server (http://www.cbs.dtu.dk/services/SignalP/ accessed on 1 September 2022). No signal peptide-triggered protein secretion was predicted by the use of the SecretomeP 2.0 server (http://www.cbs.dtu.dk/services/SecretomeP/ accessed on 1 September 2022). Prediction of virulence factors for secreted proteins was performed using VFDB (http://www.mgc.ac.cn/VFs/ accessed on 1 September 2022). After obtaining amino sequences from the NCBI, each protein was aligned separately against the VFDB full dataset by the BLAST algorithm. A matrix was created by VFDB output consisting of the BLAST score and E-value for each input protein. The data were screened on the basis of a BLAST score of ≥80. In addition, COG functional annotation for the proteins identified was acquired using the EggNOG database version 5.0 (http://eggnog5.embl.de/#/app/seqscan accessed on 1 September 2022). Furthermore, MolliGen 3.0 (http://services.cbib.u-bordeaux.fr/molligen/ accessed on 1 September 2022) was applied online to analyze the homologous proteins between *M. bovis* HB0801 and *Mmm* Afadé. The homologous proteins of Ts0085 and Ts1133 were identified from NCBI, then the multiple sequence alignment was performed by ESPript 3.0 (https://espript.ibcp.fr/ESPript/ESPript/index.php accessed on 1 September 2022).

### 2.5. Gene Cloning and Expression of Mmm-Secreted Proteins

The sequences of genes were site-direct edited by replacing the TGA codon with TGG to ensure that *Mycoplasma* tryptophan was correctly translated in *Escherichia coli* (*E. coli*). The sequence-encoded Ts0029, Ts0085, Ts0484, Ts0707, and Ts1133 were synthesized by Beijing Tianyi Huiyuan Bioscience & Technology Inc. (Wuhan, China) and ligated into the pET-30a vector (Novagen, Darmstadt, Germany). *E. coli* strain BL21 (TransGen, Beijing, China) was then transformed with each of the constructed recombinant plasmids individually, and the recombinant proteins were expressed after the *E. coli* cells were induced with isopropyl β-D-1-thiogalactopyranoside (IPTG) (0.8 mM). The proteins were purified with nickel affinity chromatography (GE Healthcare, NJ, USA), as described previously [14].

### 2.6. Characterization of the Antigenicity of Mmm-Secreted Proteins

Then, 1 μg of each purified recombinant protein (Ts1133, Ts0029, Ts0707, Ts0085, and Ts0484) was separated by 12% SDS-PAGE and transferred onto PVDF membranes (Immun-Blot^®^, USA). After blocking, recombinant proteins were probed with five serum samples of calves naturally infected with *Mmm,* kindly provided by Prof. Jiuqing Xin from China National CBPP Reference Laboratory, Harbin Veterinary Research Institute, Chinese Academy of Agricultural Sciences [25], whereas sera from non-infected calves were used as the negative control. Then, the Western blotting assay was developed with HRP-conjugated goat anti-bovine IgG (Southern Biotech Co., USA) for 1 h and finally visualized with WesternBright™ ECL (Advansta, CA, USA) [26].

In addition, the five recombinant proteins were characterized using indirect ELISA. In brief, 96-well microtitre plates were coated overnight at 4 °C with 200 ng of each purified recombinant protein diluted in 100 μL sodium carbonate buffer (pH 9.6) and washed with PBS containing 0.05% Tween 20 (PBST). After blocking, the plates were probed for 1 h at 37 °C with sera collected from *M. bovis* and *Mmm* naturally infected calves [25,27]. After washing with PBST, the plates were incubated for 1 h at 37 °C with goat anti-bovine IgG-HRP (1:5000) (Southern Biotech Co., USA) and washed with PBST, followed by the addition of tetramethylbenzidine (TMB)/H_2_O_2_ (Wuhan Keqian Biological Co., Ltd., China) as a substrate. The reaction was stopped after 5 min, and OD values at 630 nm (OD_630_) were obtained with a microtiter plate reader (BioTek, USA).

### 2.7. Statistical Analysis

The data were expressed as means ± standard error mean (SEM). Samples are normally distributed. A Student’s *t*-test was used for a single comparison with the GraphPad Prism version 5 software (GraphPad Software, La Jolla, CA, USA). * *p* < 0.05, ** *p* < 0.01, *** *p* < 0.001 indicate statistically significant differences, while ns indicates no difference.

## 3. Results

### 3.1. Soluble Secreted Proteins of M. bovis and Mmm

The growth curves showed that both *M. bovis* and *Mmm* could grow well in the semi-defined medium (Figure 1A,B). Then, the soluble secreted proteins were extracted from culture supernatants and resolved with SDS-PAGE. The results revealed that similar SDS-PAGE protein profiles were observed for the cultural supernatants of *M. bovis*, *Mmm*, or medium (Figure 1C). 

### 3.2. Identification of Proteins of M. bovis and Mmm in the Culture Supernatant

In order to identify as many secreted proteins as possible, the LC-MS/MS analysis was performed twice, and all identified proteins were selected for analysis. As a result, 48 *Mmm* and 59 *M. bovis* proteins were identified from culture supernatants (Appendix A). Further, MolliGen 3.0 (http://services.cbib.u-bordeaux.fr/molligen/ accessed on 1 September 2022) was applied to analyze online the homologous proteins between *M. bovis* HB0801 and *Mmm* Afadé. Although 454 homologous proteins were identified (Appendix A), after comparing with the secreted proteins obtained, only 4 homologous secreted proteins were shared between *M. bovis* and *Mmm*, including hypothetical protein (Mbov_0154/TS60_0462), ribosomal protein S12 (Mbov_0678/TS60_0185), heat shock protein (Mbov_0817/TS60_0689), and ATP synthase subunit alpha (Mbov_0440/TS60_0995). Moreover, *M. bovis* and *Mmm* have 55 and 44 unique secreted proteins, respectively (Figure 2A).

### 3.3. Prediction of Secreted Pathways, Locations, and Functional Analysis

Extracellular proteins are a subject of extreme interest because of their pivotal roles in bacterial lifestyles [28]. Proteins identified within the sets of *M. bovis* and *Mmm* secreted proteins were classified according to predicted subcellular protein localization using PSORTb version 3.0.2 (https://www.psort.org/psortb/ accessed on 1 September 2022) and secretion pathways. For *M. bovis*, 25 proteins (42.4%) were described as located in the cytoplasm, 8 proteins (13.6%) as cell membrane proteins, 2 (3.4%) as extracellular, and 24 (40.6%) as unknown proteins (Figure 2D). Regarding secreted pathways, 19 proteins (32.2%) were predicted to be secreted by the classical pathway, while 18 proteins (30.5%) were predicted to be non-classically secreted, and 22 proteins (37.3%) were assigned to an undefined type of secretion (Figure 2C). For *Mmm,* 24 proteins (50%) were predicted to be localized in the cytoplasm, 2 (4.2%) as extracellular, and 22 (45.8%) as unknown (Figure 2D). In this *Mmm* secretome, classical secretion was predicted for 13 proteins (27.1%), non-classical secretion was predicted for 13 proteins (27.1%), and 22 proteins (45.8%) were assigned to an undefined secretion type (Figure 2C). These proportions of location distribution for the secreted proteins of both strains are similar, but no cell membrane-localized proteins were identified in *Mmm*. In addition, the classically secreted proteins and non-classically secreted proteins in *M. bovis* and *Mmm* accounted for 62.7% and 54.2%, respectively.

The secreted proteins of *M. bovis* and *Mmm* were further categorized according to COG, and the functional prediction was summarized in Figure 2B for both *M. bovis* and *Mmm.* The identified secreted proteins were mostly related to the “function unknown” (S; 22% and 29%, respectively). In addition, the most occupied categories were cell wall/membrane/envelope biogenesis (M; 13.6% and 8.3%, respectively), replication, recombination, and repair (L; 11.9% and 8.3%, respectively), and translation, ribosomal structure, and biogenesis (J; 8.5% and 16.7%, respectively). Metabolism-related categories were also well represented for both species, comprising 25.4% of *M. bovis* identified proteins (divided in G, F, E, C, and P) and 14.6% of *Mmm* identified proteins (divided in F, E, C, P, and H). Three functional categories related to carbohydrate transport and metabolism (five proteins), defense mechanisms (three proteins), and intracellular trafficking, secretion, and vesicular transport (one protein) were found only in *M. bovis*, while the coenzyme transport and metabolism (two proteins) category was found only in *Mmm*.

### 3.4. The Virulence-Related Factors Identified by the VFDB

In order to further study the virulence-related factors of *M. bovis* and *Mmm*, all secreted proteins were analyzed using VFDB. In the VFDB full dataset, all proteins related to known and predicted virulence-related factors were present. Nineteen secreted proteins in *M. bovis* secretome were identified as virulence-related proteins based on a BLAST score ≥ 80 (Table 1), while only four proteins were identified in *Mmm* (Table 2). The proteins with top five scores in *M. bovis* include Mbov_0016 (p48) P48 predicted lipoprotein, Mbov_0482 (eno) phosphopyruvate hydratase, Mbov_0302 (sigA/rpoV) RNA polymerase, sigma 70 subunit, RpoD family (Sigma A), and Mbov_0693 (p65). P65 lipoprotein-like protein, Mbov_0675 (SAB0023) 5′ nucleotidase (AdsA). The four virulence-related proteins in *Mmm* included TS60_0188 (tuf) translation elongation factor Tu (EF-Tu), TS60_0693 (KOX_00005) protein disaggregation chaperone (T6SS-II), TS60_0792 (argK) phaseolotoxin-insensitive ornithine carbamoyltransferase, and TS60_0995 (pscN) type III secretion system ATPase.

### 3.5. The Evaluation of Differential Diagnostic Biomarkers

In order to discover the potential secreted biomarkers for distinguishing *Mmm* infection from *M. bovis* infection, we compared 48 secreted proteins of *Mmm* in our study with previously reported proteins of EV-membranes from *Mmm* [17]. The results indicated eight common proteins, and six of them were predicted as secreted proteins by bioinformatic tools (Table 3). We further expressed and purified five secreted proteins from the eight common proteins successfully (Figure 3A). Then, the five proteins were resolved with SDS-PAGE, transferred onto membranes, and incubated with antisera from five calves infected with *Mmm*. The results indicated Ts1133 and Ts0085 displayed signals in response to five antisera, respectively, but Ts0707, Ts0484, and Ts0029 did not (Figure 3B–G). iELISA was also used to demonstrate the antigenicity of recombinant proteins. In contrast to the Western blotting assay, iELISA revealed all proteins generated antibody responses with antiserum from calves infected with *Mmm* (n = 5), but Ts0085 and Ts1133 showed a stronger reaction than other proteins. In addition, the proteins in iELISA did not react with the serum infected with *M. bovis* (n = 10) or serum from the uninfected animals (n = 11). The ratio of OD values of *Mmm*-positive serum to -negative serum OD and *M. bovis*-positive serum was calculated and evaluated for the potential proteins as diagnosis targets (Figure 4). Furthermore, multiple sequence alignments revealed Ts0085 and Ts1133 were highly conserved proteins among 18 strains of *Mmm* isolated worldwide (Appendix A).

## 4. Discussion

### 4.1. M. bovis and Mmm Secretomes Are Greatly Affected by Essential Medium Components

The secretome profiles extracted from mycoplasmal culture supernatant remain extremely challenging due to the relative underrepresentation of secreted proteins in comparison to the complex background of overrepresented serum proteins [30]. Several studies attempted to identify secreted proteins in the secretomes and extracellular vesicles of several *Mycoplasma* species by proteomic approaches, such as two-dimensional electrophoresis, liquid chromatography-tandem mass spectrometry, and label-free proteomic analyses [17,18,19,20]. Our study used a semi-defined medium without serum to remove interference and facilitate the identification of secreted proteins. This approach first made the mycoplasmas adapt to the new medium shown by the normal growth curves, which, therefore, decreased the nutritional and environmental stress on the mycoplasmas to the maximum and obtained secretomes closer to those generated by mycoplasmas under the natural state compared to the previous studies [31]. However, compared to the previous findings, much less secreted proteins were obtained in this study (59 for *M. bovis*, 48 for *Mmm*), demonstrating that the secretomes greatly varied with the inclusion of medium serum, probably through disturbance of cell metabolism, protein expression, and secretion [32]. Among our secreted proteins, 25 of the *M. bovis* secreted proteins were previously identified, and 24 were predicted to be secreted by the classical pathway or non-classical pathway, indicating the current results are reliable (Table 4). Meanwhile, we also identified 48 secreted proteins of *Mmm*, and 8 of them were identified as proteins of EV-membranes, indicating that some secreted proteins of *Mmm* may be secreted as EV proteins.

On the other hand, although we successfully excluded serum interruption of the mycoplasma secretomes, the possible contamination with intracellular proteins in the supernatant could not be completely excluded. Therefore, these secretomes should be further verified with the mycoplasmas or their extracted secretomes by some other methods, such as Western blotting assays, ELISA, and in situ localization [14]. 

### 4.2. Identification of Virulence-Associated Secreted Proteins

In order to identify the virulence-associated secreted proteins, all identified proteins in the secretome were analyzed using VFDB [33,34]. As a result, there was a big difference in the number of unique virulence-associated secreted proteins: 19 for *M. bovis* and 4 for *Mmm.* The reason remains to be investigated. Among these proteins, the MbovP016 encoded by the Mbov_0016 gene has already been identified as a virulence-related factor that was able to induce apoptosis of host cells through the endoplasmic reticulum stress-dependent signaling pathway [35]; the peptide of VSP could suppress the concanavalin A-induced proliferation of bovine lymphocytes [36]. The EF-Tu encoded by the TS60_0188 gene was previously identified as an adhesin in *M. hyopneumoniae* [37]. In addition, MbovP274 was confirmed to be a secreted protein and could significantly increase the production of IL-8, IL-12, and IFN-γ [38]. Considering the limited knowledge on secreted proteins of *Mmm* and *M. bovis*, our findings provided significant reference for further identifying potential virulence-associated factors to elucidate mycoplasmal pathogenicity.

### 4.3. Potential Diagnostic Targets for Differentiation of Mmm from M. bovis

Although China and some other countries have already eradicated CBPP, its re-emergence would be possible because the exchange activities across countries free of CBPP and African countries with epidemics of CBPP are extensively performed under the conditions of economic globalization and international trade. Therefore, the development of a rapid and accurate method to rapidly differentiate *Mmm* from *M. bovis* infection is of the utmost importance. From the expressed five *Mmm* unique proteins predicted as secreted, we showed that TS0085 and Ts1133 can react solely with *Mmm*-positive cattle sera rather than *M. bovis*-positive cattle sera and negative cattle sera. Previous studies revealed MSC519 (Ts0484) and MSC636 were the best targets to identify infected animals [39,40,41]. But, our results indicated Ts0085 and Ts1133 had higher sensitivity to diagnose *Mmm* infection than Ts0484. In addition, both Ts0085 and Ts1133 were highly conserved proteins between different known strains. Although we need to test more positive sera, especially *Mmm*-positive sera, to confirm the current findings, Ts0085 and Ts1133 already show a promising application prospect as potential biomarkers for differential diagnosis of *Mmm* and *M. bovis* infection, with greater promise than other conventional diagnostic methods such as PCR. 

## 5. Conclusions

This study identified 59 secreted proteins of *M. bovis* and 48 secreted proteins of *Mmm* grown in serum-free medium. Among them, the unique *Mmm*-secreted proteins, Ts0085 and Ts1133, might serve as biomarker candidates for diagnosis of *Mmm* infection and differentiation of *Mmm* from *M. bovis* infection.

## Figures and Tables

**Figure 1 vetsci-10-00685-f001:**
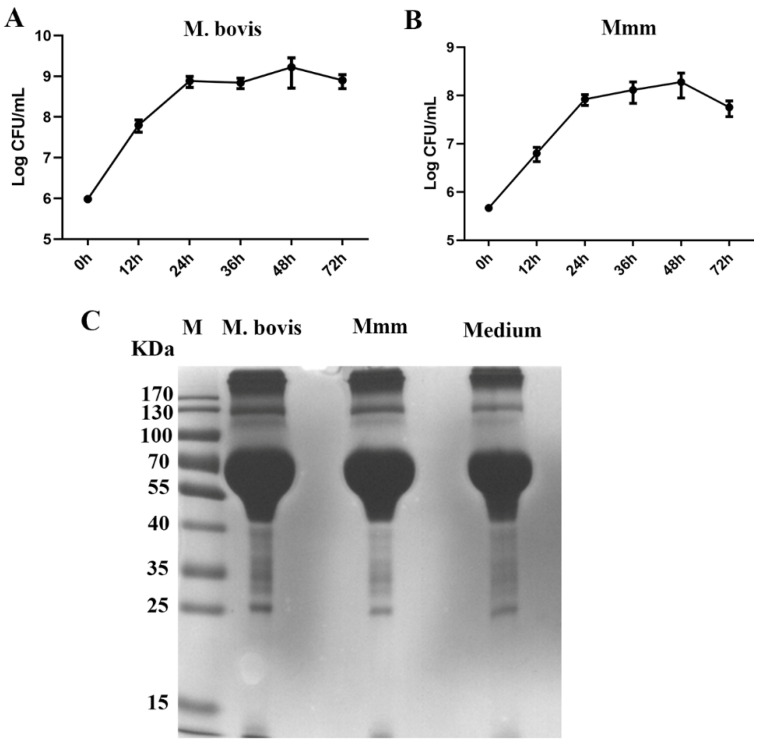
The growth of *M. bovis* and *Mmm* in the semi-defined medium without serum and their secreted proteins (three replicates each). (**A**) The growth curve of *M. bovis* in the semi-defined medium supplemented with sodium pyruvate (0.01 mol/L). (**B**) The growth curve of *Mmm* in the semi-defined medium. (**C**) The resolution of soluble secreted proteins of *M. bovis* or *Mmm* extracted from the culture supernatants with SDS-PAGE. The proteins extracted from the medium served as a control. The original gel figure can be viewed in Appendix A.

**Figure 2 vetsci-10-00685-f002:**
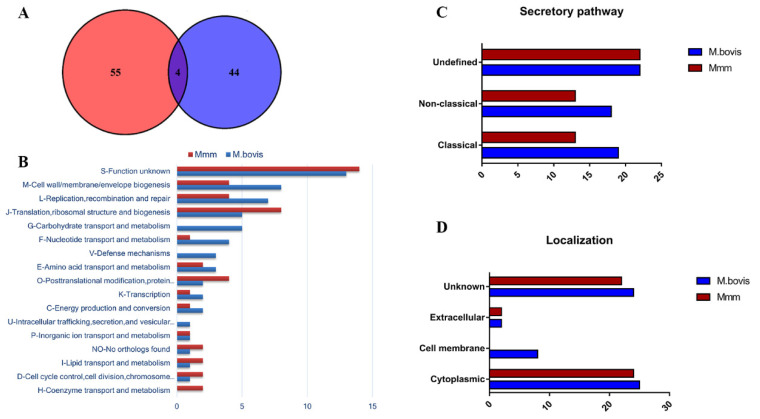
In silico functional prediction for *M. bovis* and *Mmm* secreted proteins. (**A**) The Venn diagram was created from the secreted proteins of *M. bovis* and *Mmm*. (**B**) The COG functional distribution of secreted proteins of *M. bovis* and *Mmm.* (**C**) The type of protein secretion was predicted by SignalP 5.0 and SecretomeP 2.0 servers. (**D**) The subcellular localization of secreted proteins was predicted by PSORTb version 3.0.2.

**Figure 3 vetsci-10-00685-f003:**
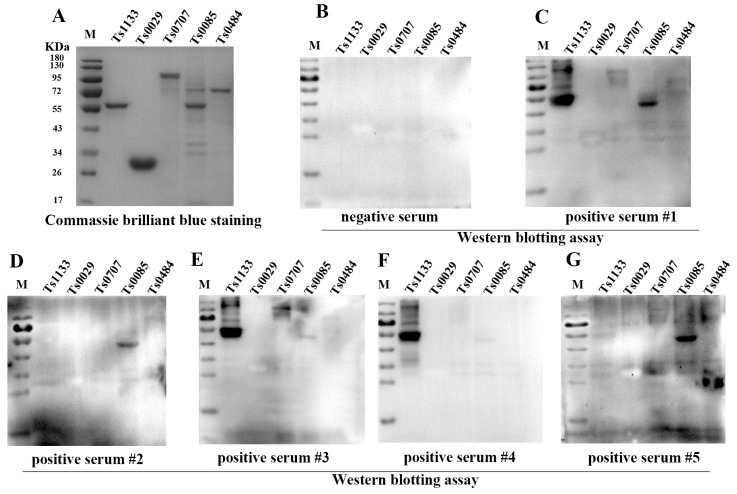
The antigenicity of 5 unique secreted proteins of *Mmm* was detected by Western blotting assay. (**A**) The SDS-PAGE of 5 unique secreted proteins of *Mmm*. The antigenicity of 5 proteins was detected with negative serum (**B**), positive serum #1 (**C**), positive serum #2 (**D**), positive serum #3 (**E**), positive serum #4 (**F**), and positive serum #5 (**G**). Then, 1 μg of each purified recombinant protein (Ts1133, Ts0029, Ts0707, Ts0085, and Ts0484) were resolved by SDS-PAGE and transferred onto PVDF membranes. Then, the antiserum (1:100) was used to detect the antigenicity [29]. The original gel figure can be viewed in Appendix A.

**Figure 4 vetsci-10-00685-f004:**
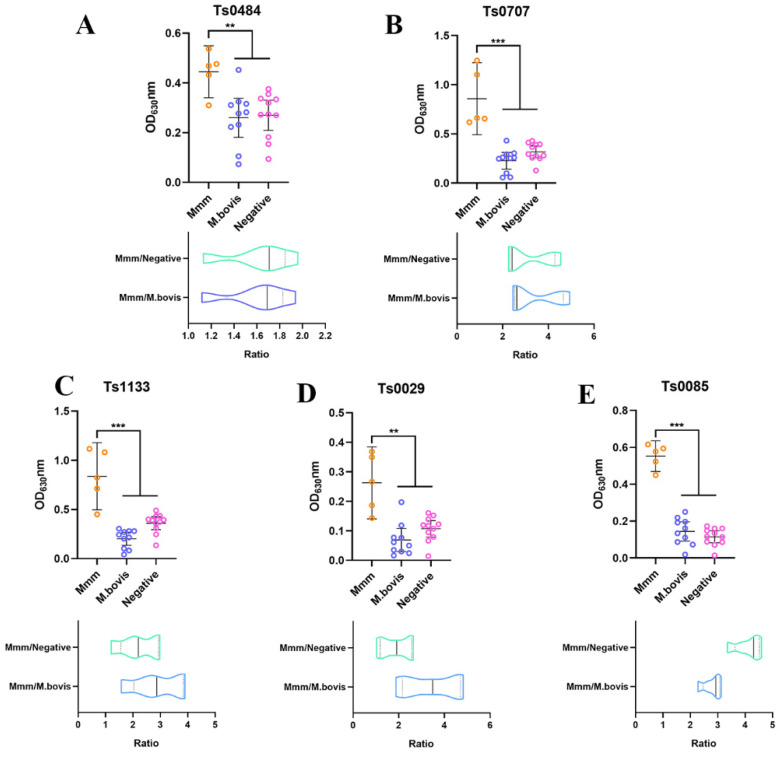
The antigenicity of 5 unique secreted proteins of *Mmm* was detected by indirect ELISA. Ts0484 (**A**), Ts0707 (**B**), Ts1133 (**C**), Ts0029 (**D**), or Ts0085 (**E**) were used to generate antibody responses with antiserum. In total, 200 ng of each protein was coated in each well overnight and incubated with antiserums of *Mmm* infection (n = 5), and *M. bovis* infection (n = 10), and negative serum (n = 11) at a dilution of 1:100 for 1 h at 37 °C. The OD_630_nm value represented antibody responses. The ratio of *Mmm*-positive serum OD to negative serum OD or *M. bovis*-positive serum OD was calculated to evaluate the ideal protein for diagnosis. ** *p* < 0.01, *** *p* < 0.001 indicate statistically significant differences.

**Table 1 vetsci-10-00685-t001:** Virulence-related factor identification in *M. bovis* using the VFDB.

Gene ID	VFDB Hits	Scores	E-Values
Mbov_0016	(p48) P48, predicted lipoprotein [Monocytic differentiation/activation factor] [*Mycoplasma agalactiae* PG2]	743	0
Mbov_0482	(eno) phosphopyruvate hydratase [Streptococcal enolase] [*Streptococcus agalactiae* A909]	443	1 × 10^−124^
Mbov_0302	(sigA/rpoV) RNA polymerase, sigma 70 subunit, RpoD family [Sigma A] [*Mycobacterium gilvum* PYR-GCK]	236	4 × 10^−62^
Mbov_0693	(p65) P65 lipoprotein-like protein [P65] [*Mycoplasma mobile* 163K]	201	8 × 10^−51^
Mbov_0675	(SAB0023) 5′ nucleotidase [AdsA] [*Staphylococcus aureus* RF122]	184	2 × 10^−46^
Mbov_0674	(nuc) MEMBRANE NUCLEASE [Nuclease] [*Mycoplasma pulmonis* UAB CTIP]	173	2 × 10^−43^
Mbov_0796	(vamp) Variable surface lipoprotein W (VpmaWprecursor) [Vpma] [*Mycoplasma agalactiae* PG2]	168	3 × 10^−42^
Mbov_0798	(vamp) Variable surface lipoprotein V (VpmaVprecursor) [Vpma] [*Mycoplasma agalactiae* PG2]	160	8 × 10^−40^
Mbov_0688	(msbA) Fused lipid transporter subunits of ABC superfamily: membrane component/ATP-binding component [LOS] [*Haemophilus influenzae* PittEE]	159	4 × 10^−39^
Mbov_0341	(p65) P65 lipoprotein-like protein [P65] [*Mycoplasma mobile* 163K]	154	9 × 10^−37^
Mbov_0440	(DVUA0119) Type III secretion system ATPase [T3SS] [*Desulfovibrio vulgaris* str. Hildenborough]	153	3 × 10^−37^
Mbov_0038	(hmw2) Predicted cytoskeletal protein [Cytadherence organella] [*Mycoplasma penetrans* HF-2]	124	1 × 10^−27^
Mbov_0134	(sugC) Maltodextrin import ATP-binding protein MsmX [Trehalose-recycling ABC transporter] [*Mycobacterium abscessus* subsp. *bolletii* str. GO 06]	122	4 × 10^−28^
Mbov_0168	(tig/ropA) trigger factor [Trigger factor] [*Streptococcus mutans* UA159]	120	2 × 10^−27^
Mbov_0509	(hrcN) HrcN [T3SS] [*Pantoea stewartii* subsp. *stewartii* str. SS104]	106	4 × 10^−23^
Mbov_0121	(cylB) ABC-type transporter [Cytolysin] [*Enterococcus faecalis* str. MMH594]	89	1 × 10^−17^
Mbov_0375	(sadH) Putative short-chain type dehydrogenase/reductase [MymA operon] [*Mycobacterium canettii* CIPT 140070010]	89	3 × 10^−18^
Mbov_0797	(vamp) Variable surface lipoprotein W (VpmaWprecursor) [Vpma] [*Mycoplasma agalactiae* PG2]	82	6 × 10^−16^
Mbov_0307	(pvuE) Iron-dicitrate transporter ATP-binding subunit [vibrioferrin] [*Vibrio parahaemolyticus* RIMD 2210633]	80	1 × 10^−15^

**Table 2 vetsci-10-00685-t002:** Virulence-related factor identification in *Mmm* using the VFDB.

Gene ID.	VFDB Hits	Scores	E-Values
TS60_0188	(tuf) Translation elongation factor Tu [EF-Tu] [*Mycoplasma mycoides* subsp. *mycoides* SC str. PG1]	789	0
TS60_0693	(KOX_00005) Protein disaggregation chaperone [T6SS-II] [*Klebsiella oxytoca* KCTC 1686]	707	0
TS60_0792	(argK) Phaseolotoxin-insensitive ornithine carbamoyltransferase [Phytotoxin Phaseolotoxin] [*Pseudomonas syringae* pv. *phaseolicola* 1448A]	147	1 × 10^−35^
TS60_0995	(pscN) Type III secretion system ATPase [*P. aeruginosa* TTSS] [*Pseudomonas aeruginosa* LESB58]	147	2 × 10^−35^

**Table 3 vetsci-10-00685-t003:** Overlapped secreted proteins of *Mmm* between our data and published data [17].

*Mmm* (ORF) ^a^	Proteins	Types of Secretion
TS60_0029	FMN-dependent NADH-azoreductase	Undefined
TS60_0085	Phosphonate ABC transporter Phosphonate-binding protein	Classical
TS60_0188	Elongation factor Tu	Undefined
TS60_0484	Hypothetical protein	Classical
TS60_0707	Hypothetical protein	Classical
TS60_0835	50S ribosomal protein L4	No-classical
TS60_0868	Hypothetical protein	Classical
TS60_1133	BspA family leucine-rich repeat Surface protein	Classical

^a^: the overlapped secreted proteins of *Mmm* between our data and secreted proteins in the EV.

**Table 4 vetsci-10-00685-t004:** Overlapped secreted proteins of *M. bovis* between our data and published data [18].

*M. bovis* (ORF) ^a^	Proteins	Types of Secretion
Mbov_0016	P48-like surface lipoprotein	Classical
Mbov_0038	Putative transmembrane protein	No-classical
Mbov_0049	Putative lipoprotein	Classical
Mbov_0106	pdhD dihydrolipoamide dehydrogenase	Undefined
Mbov_0111	Putative lipoprotein	Classical
Mbov_0154	Putative transmembrane protein	Non-Classical
Mbov_0274	Putative lipoprotein	Classical
Mbov_0290	Putative lipoprotein	Classical
Mbov_0292	vpma-like lipoprotein	Classical
Mbov_0296	Putative lipoprotein	Classical
Mbov_0341	Putative transmembrane protein	Classical
Mbov_0411	uvrA ecinuclease ABC subunit A	Non-classical
Mbov_0579	Membrane lipoprotein P81	Classical
Mbov_0674	Putative lipoprotein	Classical
Mbov_0675	cpdB 5′nucleotidase	Classical
Mbov_0678	30S ribosomal protein	Non-classical
Mbov_0688	ATP-binding cassette subfamily B	Non-classical
Mbov_0693	Putative transmembrane protein	Classical
Mbov_0760	Putative transmembrane protein	Non-classical
Mbov_0781	cdsA phosphatidate cytidylyltransferase	Non-classical
Mbov_0795	VSP	Classical
Mbov_0796	VSP	Classical
Mbov_0797	VSP	Classical
Mbov_0798	VSP	Classical
Mbov_0838	Putative lipoprotein	Classical

^a^: the overlapped secreted proteins of *M. bovis* between our data and published secretomes.

## Data Availability

The mass spectrometry proteomics data derived from this study have been deposited to iProX via the project ID IPX0007224000.

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
