# Peer review of "Comparative Proteomic Analysis of Secretory Proteins of Mycoplasma bovis and Mycoplasma mycoides subsp. mycoides Investigates Virulence and Discovers Important Diagnostic Biomarkers"

_vetsci, 2023, doi:10.3390/vetsci10120685_

Round 1

Reviewer 1 Report

Comments and Suggestions for Authors

Comments on the Quality of English Language

Moderate editing of English language required

Author Response

Thanks so much, dear reviewer for your interesting comments!

Reviewer 2 Report

Comments and Suggestions for Authors

In the present manuscript, the authors analyze and compare the secretomes of Mycoplasma bovis and Mycoplasma mycoides subsp. mycoides, which are both pathogenic in cattle. The study is of high interest as the pathogenicity of both mycoplasmas is not fully characterized, especially for Mmm.

The manuscript reads well and the materials and methods are well detailed. I just have a few comments, mostly minor, that will be hopefully considered to improve the manuscript:

Lines 32-47: M. bovis should be written in italic in the abstract.

Lines 58-59: The authors state throughout the manuscript (abstract, introduction and discussion) that Mmm has been eradicated in several countries including China. I think it would be interesting for the reader if the authors included in the introduction the name (with references) of other countries where Mmm has been successfully eradicated.

Line 102: The authors should review the text throughout the whole manuscript to correct the inconsistent use of the space before citations.

Line 116: The authors should homogenize the reference of products in the material and methods section. Sometimes the country of the company is present, sometimes not.

Line 185: How many experiments and replicates are shown in the growth curves (Figure 1A)? The authors should include this information in the legend or materials and methods.

Line 194: The authors state here that they used MolliGen 3.0 to analyze homologous protein between M. bovis and Mmm, however it is not mentioned in the materials and methods section. Please include it.

Line 203: The authors analyzed the subcellular localization of the secreted proteins using PSORTb. Even if the information is present in material and methods and in Figure 2, it would be easier for the reader to find this information also in the text in paragraph 3.3.

Line 292: The title of the paragraph should be replaced with “M. bovis and Mmm secretomes are greatly affected by essential medium components”.

Lines 335-338: To the authors knowledge, do livestock exchanges between Subsaharian countries and other countries currently take place?

Lines 338-339: How can this method outperform conventional diagnostic methods such as PCR? The authors should discuss this point, including advantages and disavantages.

Author Response

Thanks so much, dear reviewer for your informative and fruitful comments

Reviewer 3 Report

Comments and Suggestions for Authors

Review of the paper: Comparative proteomic analysis of secretory proteins of Mycoplasma bovis and mycoplasma mycoides subsp. Mycoides

The paper analysed secreted proteins of M. bovis and Mmm in order to detect which ones could serve as biomarkers. The paper found two proteins that fulfil this goal. The paper is well structured, well explained and the results are very interesting for the field. Although I found the discussion too short.

Specific comments:

L3: Mycoplasma mycoides subsp. mycoides instead of mycoplasma mycoides subsp. Mycoides

L7-19: Add the proper researchers e-mail or remove the following: ; e-mail@e-mail.com

L15-19: Correct the numbers of the addresses

L27: liquid chromatography-tandem mass spectrometry instead of C-MS/MS

L32,33, 35, 39, 42, 44, 46, 47, 48: The names of the bacteria have to be in cursive

L34: Do you mean spreaded?

L80: …mycoplasma protein pattern instead of …Mycoplasma protein pattern

L85-86: M. bovis strain HB0801 and Mmm strain Afadé, instead of Mycoplasma bovis HB0801 (M. bovis) and Mycoplasma mycoides subsp. mycoides Afadé (Mmm)

L95: strain Afadé

L98: Specify the laboratory

L102: mycoplasmas instead of Mycoplasmas

L104: mycoplasmas instead of Mycoplasmas

L111: How many microliters of PBS?

L114: How many microliters of SDS-PAGE?

L124: In silico instead of In silico

L127-138, 194: The size of the link letters is very big, reduce it.

L143: …mycoplasma tryptophan instead of …Mycoplasma tryptophan

L152: Mmm and not Mmm, because the headline is in cursive. Apply the same to all the headlines with mycoplasma names.

L154: PVDF membranes, add the brand and country

L159-160: Add information about the method or a reference.

L168: add a coma after Southern Biotech Co.

LO173: Previous to Student’s t-test it has to be demonstrated that the samples have normal distribution. If it is not the case, you have to use a non-parametric test.

L194: Information about MolliGen 3.0 should be in material and methods.

L351: secreted proteins not protein

Author Response

Thanks so much, dear reviewer for your great comments that improved the manuscript too much.

Round 2

Reviewer 1 Report

Comments and Suggestions for Authors

I have already reviewed this manuscript and wrote before: the research question is scientifically interesting. Mycoplasma bovis and Mycoplasma mycoides subsp. mycoides are concerning bovine pathogens. Many secreted proteins are virulence factors in Mycoplasma species. The study performed a comparative proteomic analysis of secreted proteins of M. bovis and M. mycoides subsp. mycoides. The results demonstrated 55 unique secreted proteins of M. bovis and 44 unique secreted proteins of M. mycoides subsp. mycoides. In both Mycoplasma secretomes, proteins were predicted as virulence factors and antigenic determinants.

As I also mentioned before, this paper presents a clear introduction as well as many experiments about secretomes of  Mycoplasma bovis and Mycoplasma mycoides subsp. mycoides. The methodology seems robust and the findings are interesting. Finally, the Discussion is brief, but the results are compared with some other important previously published studies. However, the paper had many gaps preventing the acceptance for publication.

Now the authors improved the manuscript. but there is one important pending question. So I will repeat a previous comment about this: it appears that the authors intended to investigate virulence as well as discover differential diagnostic biomarkers, as this is described in the last paragraph of the Introduction. Furthermore, Ts1133 and Ts0085 are highlighted as potential candidates for distinguishing M. mycoides infection from M. bovis infection (last sentence) in the Abstract and in a topic of the Results section (3.6. The evaluation of differential diagnostic biomarkers). However, this is not clear in the Title and Abstract. Therefore, authors need to decide whether they want to publish a descriptive article about secretomes or whether they prefer to demonstrate the different virulence factors and antigenic determinants.

In my opinion, the new title “Comparative proteomic analysis of secretory proteins of Mycoplasma bovis and Mycoplasma mycoides subsp. mycoides discovers important diagnostic biomarkers” is not mirroring the aims described in the manuscript (emphasis on virulence-related proteins). Therefore, my last suggestion to the authors is to write the title and objectives in the same logical line. It is the only pending question. Remember: the Title and the Abstract will be published in the main databases of scientific journals!

Comments on the Quality of English Language

Ok. 

Author Response

Thanks so much dear reviewer for your great comments that improved the manuscript so much!
